# Influence of Active Exposure to Tobacco Smoke on Nitric Oxide Status of Pregnant Women

**DOI:** 10.3390/ijerph15122719

**Published:** 2018-12-03

**Authors:** Magdalena Chełchowska, Jadwiga Ambroszkiewicz, Joanna Gajewska, Joanna Mazur, Leszek Lewandowski, Marzanna Reśko-Zachara, Tomasz M. Maciejewski

**Affiliations:** 1Department of Screening and Metabolic Diagnostics, Institute of Mother and Child, 01-211 Warsaw, Poland; jadwiga.ambroszkiewicz@imid.med.pl (J.A.); joanna.gajewska@imid.med.pl (J.G.); 2Department of Child and Adolescent Health, Institute of Mother and Child, 01-211 Warsaw, Poland; joanna.mazur@imid.med.pl; 3Department of Obstetrics and Gynaecology, Institute of Mother and Child, 01-211 Warsaw, Poland; leszek.lewandowski@imid.med.pl (L.L.); dyrektor.naczelny@imid.med.pl (T.M.M.); 4Department of Neonatology and Intensive Care, Institute of Mother and Child, 01-211 Warsaw, Poland; marzanna.resko@imid.med.pl

**Keywords:** tobacco smoking, pregnancy, nitric oxide, oxidative stress, endothelial nitric oxide synthase, inducible nitric oxide synthase

## Abstract

Smoking tobacco can impair proper vascular endothelial functioning. This is exhibited through reduced nitric oxide synthesis as well as activity due to accompanying oxidative stress. We examined the relationship between nitric oxide and markers of oxidative stress/antioxidant defense in serum of smoking and non-smoking pregnant women. Subjects included 99 healthy pregnant women, who were tested for nitric oxide (NO), endothelial (eNOS) and inducible (iNOS) nitric oxide synthase, total oxidant capacity (TOC), and total antioxidant capacity (TAC). NO, eNOS, and TAC serum concentrations were significantly lower (*p* < 0.005), but iNOS (*p* < 0.05) and TOC (*p* < 0.001) were higher in smokers than in non-smokers. Multivariate regression analysis showed associations between NO concentration and eNOS, TAC, and smoking status in the whole group of patients. In the model estimated separately for smokers, the highest impact of eNOS (β = 0.375; *p* = 0.021) and cotinine (β = −0.323; *p* = 0.037) was indicated for NO concentration. In the model of non-smokers, eNOS (β = 0.291, *p* = 0.030) and TAC (β = 0.350; *p* = 0.015) were important for NO level. Smoking during pregnancy could exacerbate oxidative stress, impair the action of nitric oxide synthases, and adversely affect the balance of oxygen and nitrogen metabolism. Relationships between NO concentrations and TAC in the studied women’s blood can confirm the antioxidant nature of nitric oxide.

## 1. Introduction

Smoking occupies an important place among public health problems. Poland is one of the countries where the percentage of tobacco smokers exceeds 30% of the population, including 20% of women in the reproductive period [1]. Surveys carried out at the Institute of Mother and Child in a group of 22,020 pregnant women and analysis on Polish Mother and Child Cohort conducted in a group of 1771 mothers confirmed that the percentage of women smoking during pregnancy is high and amounts to about 15–20%, and 35–50% of pregnant women are exposed to second hand smoke in the home and work environment [2,3]. Tobacco smoking is associated not only with health consequences for the mother but also with risks associated with the course of pregnancy and fetal well-being. Women who smoke during pregnancy have a higher risk of spontaneous miscarriage, ectopic pregnancy, placenta previa and premature separation, premature rupture of fetal membranes, and consequently premature birth and fetal hypotrophy. Newborns exposed to cigarette smoke in the prenatal period have a greater risk of low birth weight, decreased body length, and smaller head circumference, more frequent incidence of birth defects, as well as physical and intellectual development delays [4,5] (Table 1).

Pregnancy is characterized by intensified metabolic processes associated with the need to supply large amounts of oxygen to all the tissues and organs of the developing fetus. Reactive oxygen species (ROS) generated in both the mother and the fetus promote replication, differentiation, and maturation of the resulting cells. In the course of a normal pregnancy, increased oxidative processes connected to the intensification of lipid peroxidation and intensive production of ROS in the mitochondria, under the strict control of efficient antioxidant defense, is observed [25,26]. 

Smoking tobacco can significantly affect fetal growth in utero by an additional load of reactive oxygen species contained within, causing a disturbance in the oxidation-reduction balance [27,28,29,30]. It is estimated that one puff of tobacco smoke provides approx. 10^15^ ROS, including superoxide anion radical (O_2_^−^), hydroxyl radical (HO∙), singlet oxygen (^1^O_2_), or hydrogen peroxide (H_2_O_2_) [31].

There is a hypothesis that the harmful effects of tobacco smoke may be the result of the accumulation of oxidative endothelial cell damage. These cells synthesize endogenous nitric oxide (NO) via oxidizing L-arginine by means of nitric oxide synthases (NOS) with L-citrulline release. At least three distinct isoforms of nitric oxide synthases have been identified—two constitutive: neuronal nitric oxide synthase (nNOS, NOS1), which is primarily found in the neurons of the central and peripheral nervous system, and endothelial nitric oxide synthase (eNOS, NOS3), whose main places of expression are vascular endothelial cells, and inducible nitric oxide synthase (iNOS, NOS2), which is expressed under the influence of bacterial products, such as endotoxins and cytokines, especially in smooth muscle cells and macrophages. nNOS and eNOS are forms activated by increases in calcium concentration in the cell and the calmodulin bound by their molecules, while iNOS has such a strong affinity for the associated calmodulin that it remains active even at the lowest physiological Ca^2+^ concentrations. An important difference between the isoforms of the NO-synthesizing enzyme is that eNOS constantly produces moderate amounts of nitric oxide, whereas iNOS usually produces large amounts of NO [32,33,34].

The best-known role of nitric oxide during pregnancy is its participation in the regulation of uterine contractions and labor initiation. Nitric oxide diffuses into the myometrium and activates the heme part of soluble guanylyl cyclase to activate the smooth muscle soluble guanylyl cyclase, resulting in the accumulation of cyclic GMP (cGMP) in the muscle cells. This process stimulates the activity of cGMP-dependent kinases, proteins that cause muscles to relax. The appropriate NO concentrations inhibit the occurrence of contractile function and maintain the tonus of the uterine muscle at rest, being an element of the system protecting against the occurrence of contractile function in pregnancy before the date of delivery. Studies on the participation of this compound in the pathogenesis of preeclampsia, gestational diabetes mellitus, premature births, and gestational hypertension have been widely discussed [35,36,37]. However, the contribution of NO to the normal functioning of the vascular endothelium in the maternal-fetal unit, especially in situations of increased oxidative stress, has not been clearly defined. NO deficiency resulting from the disturbed expression of enzymes that catalyze its formation is thought to be associated with intrauterine growth restriction and low birth weight [33,38]. 

Tobacco inhaled with ROS may affect blood pressure, contributing to an imbalance between the action of vasodilating and vasoconstricting factors. Hydrogen peroxide and hydroxyl radical expand vessels, acting directly on vascular smooth muscle and stimulating NO synthesis or release; while the superoxide anion radical causes vasoconstriction, inactivating nitric oxide. The literature data describing the effect of smoking on nitric oxide levels, especially its synthases, in pregnant women are not conclusive, and relationships between NO concentration and dose and time of exposure to tobacco smoke has not been systematically investigated [38,39,40].

The aim of the study was to evaluate the effect of oxidative stress generated by smoking on the levels of nitric oxide and its synthases (inducible NOS, endothelial NOS) and their interrelations in the blood of pregnant women. To estimate the intensity of oxidative stress in the blood of the examined patients, indicators assessing Total Oxidant Capacity (TOC), Total Antioxidant Capacity (TAC), and the Oxidative Stress Index (OSI), defined as the percentage ratio of TOC to TAC level, were determined.

## 2. Materials and Methods

The test procedure was conducted according to the principles of the Declaration of Helsinki and was accepted by the Ethical Committee of the Institute of Mother and Child (Decision No. 14/2016). All pregnant volunteers were made aware of the study objectives and informed written consent was obtained for the analysis of the biological samples and linking results to the data collected from the questionnaires. 

### 2.1. Subjects

The research was carried out in a group of 99 healthy pregnant women admitted for delivery at the Department of Obstetrics and Gynecology of the Institute of Mother and Child in Warsaw in 2016–2017. A total of 42 tobacco-smoking women (admitted consecutively) were included in the study group, meeting the following inclusion criteria: gestational age over 37 weeks, single pregnancy, good health, smoking a minimum of 5 cigarettes per day throughout gestation, and smoking at least 2 years before getting pregnant. Women who had obstetric and internal problems, such as gestational diabetes mellitus, hypertension, preeclampsia, premature delivery, active hepatitis, renal and cardiovascular diseases, and those who had delivery complications and prolonged labor or assisted reproduction were not included in the study. Multiple pregnancies, declared alcohol drinking, and patients on special diets, such as vegetarian or diabetic diet, were also excluded from the study. The control group consisted of 57 women selected according to the same criteria as the study group, with the exception of smoking and passive exposure to tobacco smoke (smoking family members and/or co-workers). Qualification to the groups was confirmed by determining serum cotinine concentrations, and a cut-off value of 13.7 μg/L was considered the limit between the non-smokers and the smokers, in accordance with Jarvis et al. [41]. A medical interview was conducted with all the examined women and data on the course of pregnancy and delivery were collected, and supplemented after delivery with an assessment of the newborn’s condition. 

Anthropometric measurements, such as pre-pregnancy body height (m) and weight (kg), were performed and body mass index (BMI) was calculated as body weight (kg) divided by height squared (m^2^). Newborns were examined in the first 24 h of life. Neonatal length and weight were determined using a measuring board to the nearest 0.1 cm and a calibrated scale to the nearest 10 g. 

### 2.2. Blood Sampling and Biochemical Analysis

The material for biochemical analysis was venous blood collected (3 mL) from patients on the day of admission for delivery while performing routine examinations. The serum obtained after centrifuging the blood (2500× *g*, 4 °C, 10 min) was stored in small portions until the analysis was performed (−70 °C, less than 2 months).

Concentrations of nitric oxide, nitric oxide synthase eNOS and iNOS, and cotinine in serum were determined by immunoenzymatic methods (ELISA) using monoclonal antibodies. Nitric oxide level was evaluated using Human (NO) ELISA kit (SunRed; Shanghai, China). Levels of eNOS and iNOS were determined using the Human NOS3 ELISA and Human NOS2 ELISA sets, respectively (Cloud-Clone Corp., Houston, TX, USA). The manufacturer’s data on intra- and inter-assay coefficients of variation (CV) were less than 8.0% and 11.0% for NO, 10.0% and 12.0% for NOS, respectively. Cotinine concentration was measured using a commercially available kit, Cotinine Direct ELISA (Calbiotech Inc., Spring Valley, CA, USA). The intra- and inter-assay CV for this method were less than 10%. 

Total oxidant capacity and total antioxidant capacity values were measured by colorimetric assay based on the enzymatic reaction of peroxides and peroxidases according to Tatzber et al. [42] (Labor Diagnostica Nord GmbH&Co.KG, Nordhorn, Germany). The manufacturer’s data on intra- and inter-assay coefficients of variation were less than 4.9% and 7.33% for TOC, 2.5% and 3.33% for TAC, respectively. OSI was calculated from the measured values of TOC and TAC using the TOC/TAC ratio multiplied by 100 [43,44].

### 2.3. Statistical Analysis

The statistical analyses of the results were performed using SPSS statistical software version 17.1 (SPSS INC., Chicago, IL, USA). Normality of variables was tested using the Kolmogorov–Smirnov test. The symmetrically distributed data were presented as means with standard deviation (SD) and the asymmetrically distributed data as medians with interquartile range (25–75th percentiles). The baseline characteristics and biochemical parameters were compared using the Student *t*-test or Mann–Whitney *U* test, depending on the assumptions. The Chi-squared test was used for comparing nominal variables. Correlation analysis was performed based on Pearson or Spearman coefficients, adequately to the distribution of variables. The potential influence of oxidative stress markers on the level of nitric oxide was estimated by linear regression analysis. Results were presented as the value of B unstandardized regression coefficient with 95% confidence interval and change in R-squared coefficient after each variable was entered. Models were estimated separately for smokers and non-smokers as well as for the total group. A *p* value of <0.05 was considered statistically significant. 

## 3. Results

Pregnant women in both groups were of similar age (28.5 ± 4.5 years vs. 29.8 ± 4.2 years; *p* = 0.091, smokers and non-smokers, respectively) and did not differ in terms of gestational age (39.1 ± 1.1 weeks of pregnancy vs. 39.5 ± 1.0 weeks of pregnancy, *p* = 0.055, smokers and non-smokers, respectively). The patients’ anthropometric parameters were similar in the study and the control groups (height: 163.8 ± 4.4 cm vs. 165.5 ± 4.6 cm, *p* = 0.075; weight: 63.7 ± 5.5 kg vs. 65.3 ± 6.1 kg, *p* = 0.187; BMI: 23.7 ± 1.6 vs. 23.8, *p* = 0.804, respectively). In the group of smokers, the median value of number of cigarettes smoked per day was 8 (mean: 8.86 ± 4.29; range: 5–20), and the median of duration of the habit before conception was 8 years (mean: 8.26 ± 3.39; range: 2–15). In this group, blood cotinine levels ranged from 51.4 μg/L to 107.2 μg/L, while in the tobacco abstinent group cotinine was found in only two cases in trace amounts (0.0 μg/L to 2.1 μg/L). The newborns of actively smoking mothers did not differ in terms of sex and Apgar score from the newborns of non-smokers (42.9% girls, 57.1% boys, 10 points vs. 43.9% girls, 56.1% boys, 10 points, *p* = 0.543, respectively). The birth weight and length of the children of smokers was significantly lower than those of non-smokers (3088 ± 407 g vs. 3521 ± 397 g, 53.9 ± 2.5 cm vs. 56.0 ± 2.5 cm, respectively, *p* ≤ 0.001). 

NO, eNOS, and TAC concentrations in the serum of smokers were significantly lower than those observed in the tobacco abstinent group. In addition, iNOS, TOC, and oxidative stress index values were significantly higher than in the group of non-smokers (Table 2). In smokers, the nitric oxide level was negatively correlated with the number of cigarettes smoked per day and cotinine concentration (r = −0.516, *p* ≤ 0.001). A similar relationship occurred between eNOS level and cotinine concentration (r = −0.334, *p* = 0.031) while the correlation of eNOS with the number of cigarettes smoked per day was on the border of statistical significance (r = −0.294, *p* = 0.059). We found no significant relationships between iNOS level and the studied parameters describing smoking intensity.

Based on regression analysis, a positive correlation between NO concentration and eNOS level and a negative relationship between NO and iNOS concentrations was found in the whole group of patients. Close correlations between nitric oxide and endothelial form of nitric oxide synthase were confirmed in both the study and the control groups (Table 3).

Table 4 and Figure 1, Figure 2, Figure 3, Figure 4 and Figure 5 present relations between the studied nitric oxide parameters and oxidative stress markers separately for the smokers and the non-smokers groups. There was a significant positive relationship between NO and TAC in both studied groups, while an inverse association between NO and TOC was found only in the smoking group (Figure 1 and Figure 2). 

Serum concentration of NO slightly correlated negatively with the OSI value in smoking and non-smoking mothers, but it was on the border of statistical significance (Figure 3).

The level of serum eNOS correlated positively with TAC in the non-smoking group and negatively with TOC in the smoking group (Figure 4 and Figure 5). 

There was no significant association between eNOS and OSI in both studied groups (Table 3). Positive associations between iNOS concentrations and TOC as well as OSI values in the group of smoking mothers were found. We did not observe connections between iNOS and TAC levels in any of the studied groups (Table 4).

In the multivariate regression model (adjusted for age, gestational age, BMI), we observed associations between nitric oxide concentration and eNOS, TAC, and smoking status in the whole group of pregnant women (Table 5). 

In the model estimated separately for smokers, the highest impact of the serum eNOS and concentration of cotinine was indicated for NO concentration. In the model of non-smokers, eNOS and TAC were of significant importance for the nitric oxide level. R-squared (expressed as percentage of a variation that can be explained by a linear regression model) was 54.1 for the whole group of women, 45.2 for the smokers group, and 32.7 for the tobacco abstinent group.

## 4. Discussion

In the presented study, we found significant correlations between serum concentration of nitric oxide and its synthases, and oxidative stress markers in pregnant women. We also confirmed a negative effect of tobacco smoking on the studied biochemical parameters. 

It is widely known that smoking intensifies free radical generation processes, often coexisting with an insufficiency of antioxidant systems, manifested by a decline in the concentrations of individual antioxidants and damage to the endothelium of blood vessels [30,45,46]. This results in impaired synthesis of endogenous nitric oxide, which in turn can lead to dangerous thickening and stiffening of blood vessel walls. An imbalance between the action of vasodilating and vasoconstricting factors may be the result of a loss of diastolic capacity dependent on nitric oxide, as well as a reduction in NO production or availability in both uteroplacental and fetal-placental circulation. Trophoblast-derived nitric oxide may prevent the adhesion and aggregation of platelets and leukocytes in syncytiotrophoblasts, maintaining proper circulation in the intervillous space [33,38,45,47]. 

The presented study shows that smoking tobacco significantly reduces nitric oxide levels in the blood serum of pregnant women, which is consistent with our previous reports [40] and the results obtained by other authors [31,38,39]. Similarly to the study by Node et al. [48], the level of this compound was dependent on the amount of cigarettes smoked. The close positive correlation between NO and total antioxidant capacity we observed in both groups may confirm the antioxidative nature of this compound suggested by other authors [32,49]. Negative correlations between NO and total oxidant capacity in the blood of smoking women coexisting with lower TAC levels in this group seem to explain the influence of oxidative stress caused by smoking on nitric oxide concentration. This may be due to disturbed expression and the associated reduction in the amount of the eNOS form. It was shown that ROS from tobacco smoke induced eNOS uncoupling secondary by reducing the concentration of tetrahydrobiopterin (BH4), which is an important cofactor for the production of NO with NOS susceptible to oxidative stress [32]. People who smoke also have disorders related to the cyclooxygenase (COX) pathway, which might be involved in NO homeostasis disturbances [50,51]. An in vitro study shown that an increase in intracellular O_2_ levels observed in histone-treated HUVEC, at least in part produced by COX-2 activity, contributes to decreased eNOS expression and NO bioavailability [52]. 

There are few data in the literature describing the effect of smoking on nitric oxide synthase concentrations in pregnant women. The study by Andersen et al. [33,38] indicates that eNOS activity and concentration in umbilical cord endothelial cells were significantly lower in the children of smokers than children of non-smokers. As a result of multivariate regression analysis, these authors found that a 20% reduction in eNOS activity, associated with smoking, may be due to reduced eNOS concentrations, and 5% due to reduced HDL concentrations [38]. It is known that tobacco smoking disturbs the lipid profile in pregnant women, who have higher levels of LDL and lower HDL than non-smokers, which predisposes this group to oxidative lipid damage and subsequent damage to cell membranes, including endothelial cells [28,53]. 

In our study, eNOS concentrations in the serum of smoking pregnant women were lower than in the tobacco abstinent group. In addition, nitric oxide levels were strictly dependent on endothelial nitric oxide synthase levels in both studied groups. This may be the result of lower estrogen and L-arginine levels in smokers, which leads to a rise in asymmetric dimethylarginine (ADMA), an inhibitor of nitric oxide synthase [54,55]. A decrease of about 30% in estriol after 37 weeks of pregnancy has been observed in women smokers, which can directly affect endothelial NO synthase concentration and nitric oxide bioavailability [33]. 

The effect of oxidative disturbances on estrogen levels and eNOS regulation has been observed in both animal and human studies [56,57,58]. Novella et al. [59] found that oxLDL increased ADMA levels in human arterial endothelial cells. The authors suggest that estradiol may protect cells from oxLDL-induced damage. This could involve a decline in ADMA levels and an enhancement of NO release. It is known that oxidative stress contributes to endothelial dysfunction in estrogen-deficient postmenopausal women. An in vitro study conducted in mice with experimental menopause shown that the interaction between NOS and COX pathways is altered by aging as well as estrogen status and is driven partially by COX-mediated generation of superoxide (O_2_^−^), which deceases NO bioavailability [60]. The negative correlations we observed between eNOS and cotinine and TOC seem to confirm the involvement of oxidative stress in the formation and activity of this enzyme in pregnant women who smoke tobacco.

The nicotine contained in tobacco smoke affects the endothelium, leading not only to permanent damage but also to reducing nitric oxide synthesis. In vivo studies showed that local exposure to nicotine is associated with a decreased response to bradykinin in human vasculature, suggesting that vasoconstriction may be the result of inhibited eNOS activity [61]. The hypoxia resulting from the action of carbon monoxide also lowers both eNOS activity and concentration in human umbilical cord endothelial cells compared with conditions with normal oxidation status [62]. 

The increase in reactive oxygen species (ROS) production observed in smoking pregnant women and the reduced possibility of their inactivation by antioxidant mechanisms may reduce NO synthesis and decrease NO availability [27,29,63]. It is highly probable that higher inducible nitric oxide synthase activity and concentration results from ROS generation by tobacco smoke. In our studies, iNOS level was significantly higher in women smokers compared with non-smokers and was positively correlated with OSI and TOC levels in the blood. This is concordant with the results of other studies conducted in animals and in human epithelial endometrial cell cultures [34]. In vitro experiments showed that exogenous nitric oxide may increase iNOS expression in macrophages stimulated by interferon gamma and lipopolysaccharides [49]. Chronic inflammation, often accompanying smoking, may be a source of pro-inflammatory cytokines and thus induce iNOS expression [53,64]. 

Due to NO presence in tobacco smoke (about 500 ppm of NO and other active nitrogen oxides), it was expected that as a result of smoking, the NO metabolites in systemic circulation will increase [65]. Animal studies suggest that inhalation of tobacco smoke causes an increase in NO concentration in the plasma, which is probably the result of the release of exogenous NO from tobacco smoke. In the animal model, exposure to tobacco smoke was much higher than the average tobacco smoker is exposed to, which may explain the high levels of this compound found in the examined animals [66]. Chambers et al. [67] showed a temporary increase in NO concentrations in smokers. However, it seems that the temporary increase in NO concentration results from the inhalation of exogenous NO contained in smoke and refers only to the respiratory tract of active smokers. It can also be assumed that the long-term reduction in eNOS activity in people who are actively smoking has a greater influence on the status of endogenous NO in these people than transient exposure to exogenous NO from tobacco smoke.

An important limitation of the present study was the small sample size; however, the studied group was homogeneous. All women were similar in age, age of gestation, and type of delivery, which are recognized as important factors for nitric oxide and oxidative stress marker levels [68]. Another limitation is that we did not use a typical study design for a case-control study with retrospective assumptions, and a prospective comparison of vulnerable groups and those not exposed to one factor, in this case tobacco smoking. Although the recruitment method used did not allow reaching the percentage of smokers representing a normal pregnant population, we managed to compare groups that did not differ in size or basic characteristics [69]. We lacked data on inflammatory marker concentrations (except for CRP) in the studied patients, but all women were healthy without inflammation processes confirmed by negative C-reactive protein. Also, we did not have data on estrogen levels, but it was clear from the medical history that no patient received estrogen treatment during pregnancy. There were no complications in the course of pregnancy, and all births were through natural passage at full term. Although it is known that smoking markedly affects antioxidant estrogen status, determining the relations between them and oxidative stress markers generated by smoking in pregnancy should be continued [57,58].

The presented results can be used to establish rational supplementation with preparations containing antioxidants for the prevention of oxidative/antioxidant balance disorders and as a consequence of free radical damage and related complications.

In summary, the obtained results indicate that smoking during pregnancy could exacerbate oxidative stress, impair the action of nitric oxide synthases, and adversely affect the balance of oxygen and nitrogen metabolism. Relationships between NO concentrations and total antioxidant capacity in the studied women’s blood can confirm the antioxidant nature of nitric oxide.

## Figures and Tables

**Figure 1 ijerph-15-02719-f001:**
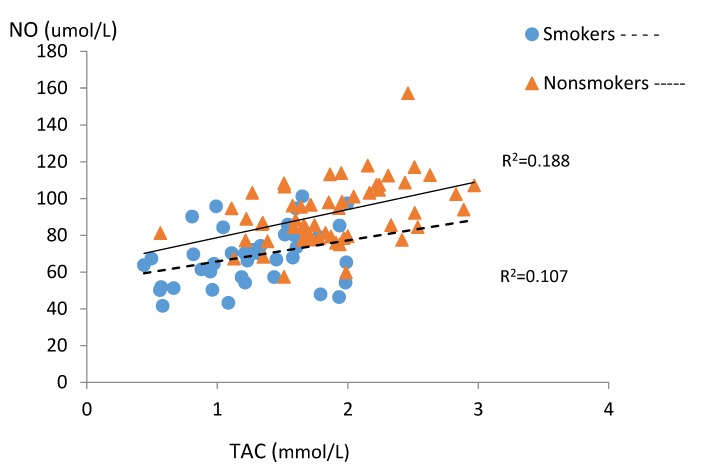
Correlations between serum NO and TAC levels in the subgroups of smoking (n = 42) and non-smoking pregnant women (n = 57).

**Figure 2 ijerph-15-02719-f002:**
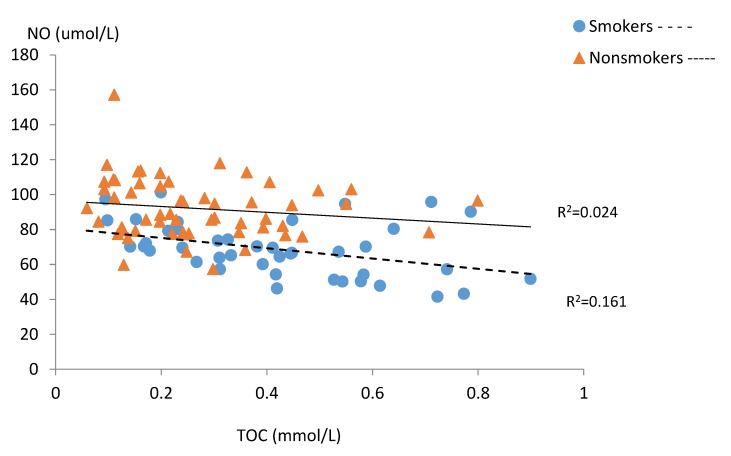
Correlations between serum NO and TOC levels in the subgroups of smoking (n = 42) and non-smoking pregnant women (n = 57).

**Figure 3 ijerph-15-02719-f003:**
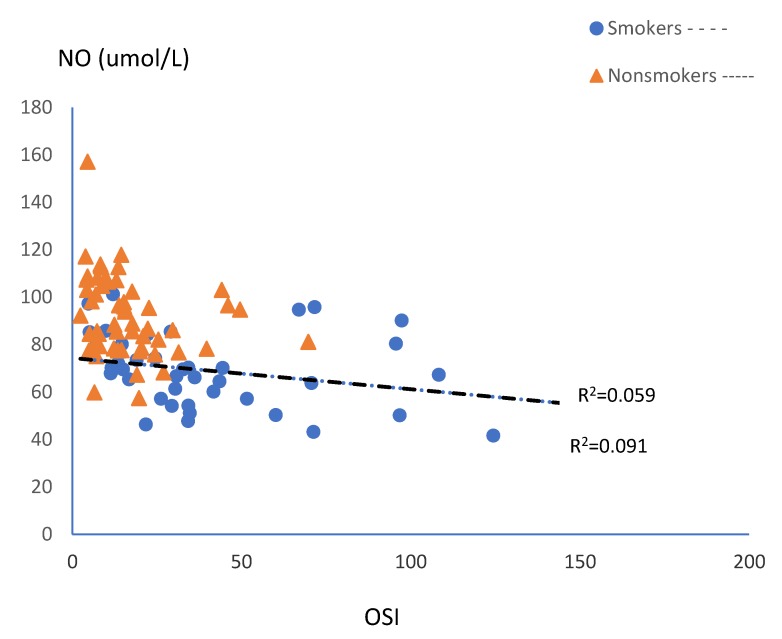
Correlations between serum NO and OSI levels in the subgroups of smoking (n = 42) and non-smoking pregnant women (n = 57).

**Figure 4 ijerph-15-02719-f004:**
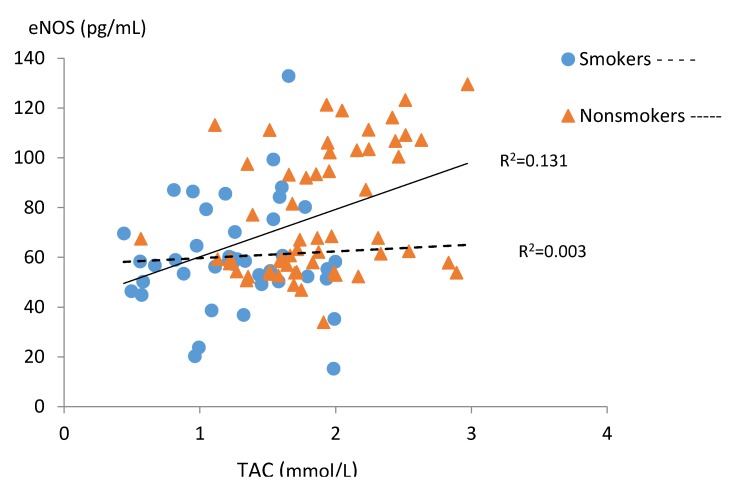
Correlations between serum eNOS and TAC levels in the subgroups of smoking (n = 42) and non-smoking pregnant women (n = 57).

**Figure 5 ijerph-15-02719-f005:**
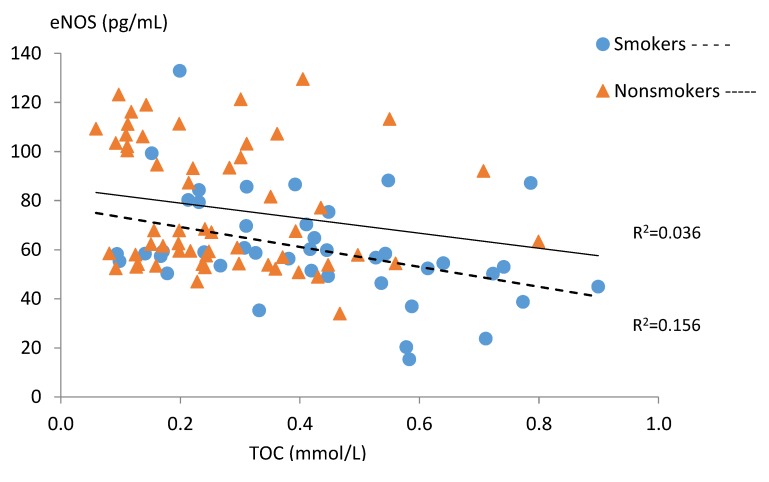
Correlations between serum eNOS and TOC levels in the subgroups of smoking (n = 42) and non-smoking pregnant women (n = 57).

**Table 1 ijerph-15-02719-t001:** Literature review on the adverse effect of smoking on the progress of pregnancy and the fetus.

Newborn	References
Low birth weight	[6]
Stillbirth	[7,8]
Congenital malformation	[9]
Increased risk of sudden infant death syndrome (SIDS)	[10]
Behavior disorders	[11]
Neurodevelopmental disorders	[12]
Increased risk of overweight and obesity	[13]
Increased risk of respiratory allergy	[14]
Higher blood pressure	[15]
Increased risk of gestational diabetes	[16]
**Pregnant women**	
Preterm premature rupture of membranes	[17]
Placenta previa	[18]
Ectopic pregnancy	[19]
Spontaneous miscarriage	[20]
Preterm birth	[21,22]
Fetal growth restriction	[23]
Fetal size	[24]

**Table 2 ijerph-15-02719-t002:** Serum concentration of biochemical parameters in the subgroups of smoking (n = 42) and non-smoking pregnant women (n = 57).

Biochemical Parameter	Smoking	Non-Smoking	*p* Value
NO (µmol/L) ^a^	68.8 ± 15.4	92.1 ± 16.9	≤0.001
eNOS (pg/mL) ^a^	60.4 ± 21.6	76.9 ± 25.5	0.003
iNOS (pg/mL) ^a^	101.5 ± 21.0	92.3 ± 19.3	0.020
TAC (mmol/L) ^a^	1.265 ± 0.441	1.877 ± 0.483	≤0.001
TOC (mmol/L) ^a^	0.417 ± 0.210	0.267 ± 0.157	≤0.001
OSI ^b^	42.3 (14.8–61.8)	13.6 (7.2–20.1)	≤0.001

Data are presented as ^a^ mean values ± standard deviation (SD), ^b^ median values and interquartile ranges (25–75th); NO—nitric oxide, eNOS—endothelial nitric oxide synthase, iNOS—inducible nitric oxide synthase, TOC—total oxidant capacity, TAC—total antioxidant capacity, OSI—oxidative stress index.

**Table 3 ijerph-15-02719-t003:** Univariable regression analyses of NO with their synthases eNOS and iNOS in the whole group of studied women (n = 99) and in the subgroups of smoking (n = 42) and non-smoking pregnant women (n = 57).

Biochemical Parameter	B	95%CI	β	*p*-Value	R^2^
All women					
eNOS	0.431	0.297/0.565	0.543	≤0.001	0.295
iNOS	−0.244	−0.434/−0.053	−0.249	0.013	0.062
Smokers					
eNOS	0.353	0.155/0.552	0.494	0.001	0.244
iNOS	−0.152	−0.382/0.078	−0.207	0.189	0.043
Non-smokers					
eNOS	0.294	0.133/0.456	0.441	0.001	0.195
iNOS	−0.099	−0.335/0.138	−0.112	0.407	0.013

NO—nitric oxide, eNOS—endothelial nitric oxide synthase, iNOS—inducible nitric oxide synthase.

**Table 4 ijerph-15-02719-t004:** Relations between studied nitric oxide parameters and oxidative stress markers in the subgroups of smoking (n = 42) and non-smoking pregnant women (n = 57).

	Smoking						Non-Smoking					
	NO		eNOS		iNOS		NO		eNOS		iNOS	
	β	*p*-Value	β	*p*-Value	β	*p*-Value	β	*p*-Value	β	*p*-Value	β	*p*-Value
TAC	0.327	0.033	0.056	0.994	−0.197	0.211	0.435	0.001	0.361	0.006	0.055	0.684
TOC	−0.401	0.008	−0.395	0.010	0.472	0.002	−0.155	0.250	−0.190	0.158	0.239	0.074
OSI	−0.301	0.053	−0.237	0.131	0.410	0.007	−0.243	0.069	−0.210	0.117	0.119	0.377

NO—nitric oxide, eNOS—endothelial nitric oxide synthase, iNOS—inducible nitric oxide synthase, TOC—total oxidant capacity, TAC—total antioxidant capacity, OSI—oxidative stress index.

**Table 5 ijerph-15-02719-t005:** Multivariable regression analyses (adjusted for age, gestational age, BMI) of NO with eNOS iNOS, markers of oxidative stress and smoking status in the whole group of studied women (n = 99), smoking women (n = 42), and tobacco abstinent group (n = 57).

Biochemical Parameter	B	95%CI	β	*p*-Value
All women eNOS	0.268	0.141/0.395	0.338	≤0.001
iNOS	−0.075	−0.228/0.079	−0.076	0.338
TAC	6.421	−0.400/13.242	0.178	0.065
TOC	−5.359	−23.922/13.204	−0.052	0.568
Smoking status (no = 1, yes = 0)	−12.299	−19.340/−5.257	−0.306	0.001
R^2^ (%)	54.1			
Smokers				
eNOS	0.275	0.045/0.505	0.383	0.021
iNOS	−0.076	−0.313/0.161	−0.103	0.519
TAC	3.006	−8.338/14.350	0.086	0.593
TOC	−3.746	−33.082/25.590	−0.051	0.797
Cotynine	−0.152	−0.295/−0.009	−0.323	0.037
R^2^ (%)	45.2			
Non-smokers				
eNOS	0.194	0.020/0.368	0.291	0.030
iNOS	−0.100	−0.323/0.123	−0.113	0.373
TAC	12.282	2.506/22.057	0.350	0.015
TOC	5.434	−23.474/34.342	−0.164	0.417
R^2^ (%)	32.7			

NO—nitric oxide, eNOS—endothelial nitric oxide synthase, iNOS—inducible nitric oxide synthase, TOC—total oxidant capacity, TAC—total antioxidant capacity, OSI—oxidative stress index.

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
