# Peer review of "Influence of Active Exposure to Tobacco Smoke on Nitric Oxide Status of Pregnant Women"

_ijerph, 2018, doi:10.3390/ijerph15122719_

Round 1

Reviewer 1 Report

IJERPH 387136 review

Overview: “The aim of the study was to evaluate the effect of oxidative stress generated by smoking on the levels of nitric oxide and its synthases (inducible NOS, endothelial NOS) and their interrelations in the blood of pregnant women.” 99 pregnant women had routinely collected late pregnancy blood samples analyzed for nitrous oxide, iNOS eNOS, TOC, TAC, and TOC/TAC as a measure of oxidative stress (OSI). Comparisons were made between smoking and non-smoking women. NO, eNOS, and TAC were suppressed in smokers, while iNOS and TOC were elevated.

Overall, the paper is very well written and should be published after some clarifications and probably some additional analyses.

Introduction

Some of the wording is unusual. On line 36, smokers might dispute that their habit is among the “abnormalities pertaining to life style.” That it is a risk factor for numerous diseases would not be in dispute. On line 42, “nicotinism” covers a great many behaviors, including smoking, chewing, dipping, and sniffing tobacco; NRT; e-cigs; and all novel nicotine delivery systems. Of these, smoking is by far the most dangerous and best documented risk for pregnancy outcomes. Nicotinism does not cover second hand smoking, which probably poses some risk. I would suggest replacing “nicotinism” with “smoking.”

Lines 74-81 would benefit from a few more specific words describing the effects of NO on these processes, e.g. that NO contributes to cervical thinning and uterine relaxation.

Line 93 or 138, OSI needs a reference and some indication of normal or expected values. Here’s one that seems not to pertain at all:https://innovaticslabs.com/wp-content/uploads/2018/04/OSI_Oxidative-Stress-Index.pdf

Line 133, in the methods, the coefficients of variation are presumably reported by Labor Diagnostica Nord? Some phrase to this effect would be helpful. The current sentence makes it sound like the CV might have been calculated from the data. If these were calculated from data in the study, rather than reported by the lab, then probably this should be in the results.

Methods: If the aim is “to evaluate the effect of oxidative stress generated by smoking on the levels of nitric oxide and its synthases” and the oxidative stress is likely to be smoking dose-responsive, why not start by demonstrating how a range of smoking doses are correlated with OSI, and then show how OSI correlated with NO and synthases (and perhaps even birth weight)? The smoking dose could reasonably be estimated as cigarettes per day and again as serum cotinine (although this is dependent on genetics to some degree) or perhaps CO. The authors have CPD and cotinine measures, but dichotomized the sample to a 0/1 measure of smoking status. Unless the CPD and cotinine are very tightly grouped (SD’s are not given in the text), the dichotomization of smoking status probably loses information by treating all smokers as receiving the same dose. Also, the paper sort of bogs down presenting correlations of TAC and TOC while failing to make thorough use of the OSI. If the authors present the data and analysis this way because OSI does not correlate with CPD or NO, then we really deserve to know that this is not dose-responsive, but an on-off switch related to smoke exposure (at least at these levels of smoking).

In figures 1-4, the R2lines ought to be consistently marked as smoking or non-smoking, perhaps by making one of the lines consistently dashed, color coding, or labeling the line formula (R2S= …, R2N=…). 

Line 292-4, “In vivo studies showed that local exposure to nicotine is associated with decreased response resulting from the endothelial action of NO in human vasculature, suggesting that vasoconstriction may be the result of inhibited eNOS activity [40].” The report [40] says “nicotine administration was associated with a loss in sensitivity to bradykinin,” even in the presence of indomethacin that would block prostacyclin synthesis: this is suggestive but does not definitively relate the response to NO inhibition, because there is still a possibility of a third vasodilator, EDHF. A more conservative sentence might be, “In vivo studies showed that local exposure to nicotine is associated with decreased response to bradykinin in human vasculature, suggesting that vasoconstriction may be the result of inhibited eNOS activity [40].”

Line 300-1 is not clear to me: “An increase in inducible nitric oxide synthase activity and concentration may be a secondary process for ROS generation by tobacco smoke.” Does this mean that higher iNOS causes more ROS generation? Is it including NO as an ROS? Or should it say that higher iNOS results from ROS generation by tobacco smoke?

Line 309-317: I would have supposed that the (fairly long lasting) decline in eNOS more than offset the (fairly transient) exogenous NO exposure plus iNOS activity.

Author Response

Review Report Form 1

Overview: “The aim of the study was to evaluate the effect of oxidative stress generated by smoking on the levels of nitric oxide and its synthases (inducible NOS, endothelial NOS) and their interrelations in the blood of pregnant women.” 99 pregnant women had routinely collected late pregnancy blood samples analyzed for nitrous oxide, iNOS eNOS, TOC, TAC, and TOC/TAC as a measure of oxidative stress (OSI). Comparisons were made between smoking and non-smoking women. NO, eNOS, and TAC were suppressed in smokers, while iNOS and TOC were elevated.

Overall, the paper is very well written and should be published after some clarifications and probably some additional analyses.

Introduction

Some of the wording is unusual. On line 36, smokers might dispute that their habit is among the “abnormalities pertaining to life style.” That it is a risk factor for numerous diseases would not be in dispute. On line 42, “nicotinism” covers a great many behaviors, including smoking, chewing, dipping, and sniffing tobacco; NRT; e-cigs; and all novel nicotine delivery systems. Of these, smoking is by far the most dangerous and best documented risk for pregnancy outcomes. Nicotinism does not cover second hand smoking, which probably poses some risk. I would suggest replacing “nicotinism” with “smoking

Corrected as suggested (line 37, line 43).

Lines 74-81 would benefit from a few more specific words describing the effects of NO on these processes, e.g. that NO contributes to cervical thinning and uterine relaxation.

According to the Reviewer’s recommendation, we added some information about the role of NO in uterine relaxation (line 80-86).

Line 93 or 138, OSI needs a reference and some indication of normal or expected values. Here’s one that seems not to pertain at all:https://innovaticslabs.com/wp-content/uploads/2018/04/OSI_Oxidative-Stress-Index.pdf

Corrected as suggested. In our study, TOC and TAC determination was done using a different manufacturer’s sets than that used to calculate OSI (all:https://innovaticslabs.com/wp-content/uploads/2018/04/OSI_Oxidative-Stress-Index.pdf).  Most often, Oxidative Stress Index (OSI) is defined as the percentage ratio of TOC to TAC level, which was taken into account in our study. We added references in the literature pertaining to OSI values observed by other authors in pregnant women as well as in Polish children (line 104-105; line 153-154; line 443-448).

Line 133, in the methods, the coefficients of variation are presumably reported by Labor Diagnostica Nord? Some phrase to this effect would be helpful. The current sentence makes it sound like the CV might have been calculated from the data. If these were calculated from data in the study, rather than reported by the lab, then probably this should be in the results.

Information in line 133 (previously) refer to CV for NO, eNOS, iNOS and cotinine. All values of coefficients of variation are the values provided by the set manufacturers. According to the reviewer's suggestion, the necessary information was included in the text (line 144-145; line 147-148; line 151).

Methods: If the aim is “to evaluate the effect of oxidative stress generated by smoking on the levels of nitric oxide and its synthases” and the oxidative stress is likely to be smoking dose-responsive, why not start by demonstrating how a range of smoking doses are correlated with OSI, and then show how OSI correlated with NO and synthases (and perhaps even birth weight)? The smoking dose could reasonably be estimated as cigarettes per day and again as serum cotinine (although this is dependent on genetics to some degree) or perhaps CO. The authors have CPD and cotinine measures, but dichotomized the sample to a 0/1 measure of smoking status. Unless the CPD and cotinine are very tightly grouped (SD’s are not given in the text), the dichotomization of smoking status probably loses information by treating all smokers as receiving the same dose. Also, the paper sort of bogs down presenting correlations of TAC and TOC while failing to make thorough use of the OSI. If the authors present the data and analysis this way because OSI does not correlate with CPD or NO, then we really deserve to know that this is not dose-responsive, but an on-off switch related to smoke exposure (at least at these levels of smoking).

We are very grateful to the Reviewer for drawing our attention to this issue. Detailed data on the value of markers assessing smoking intensity has been added to the results (line 175-177). Determination of blood cotinine levels is considered a very good marker for assessing smoking intensity its value in our study was closely correlated with the number of cigarettes smoked per day (r=0.897, p<0.001). Therefore, serum concentration of cotinine has been added as an additional factor in the multivariate regression model specific for smokers. The inclusion of this factor to the model caused that the R-squared for this model increased to 45% (actually: Table 5).

The authors presented the correlation between NO and its synthases and TOC and TAC first, because these are the parameters determined in patients’ blood that illustrate the intensity of oxidation processes and the efficiency of antioxidant defense and are described by other authors in studies on pregnant women. In our study, TOC levels in smokers were higher than in non-smokers and correlated with both the number of cigarettes smoked per day (r=0.351, p=0.023) and cotinine level (r=0.320, p=0039). TAC concentration was significantly lower in the smokers group but did not show any significant connections with CDP (r=-0.170, p=0.282) and cotinine (r=-0.080, p=0.959). Because OSI is calculated from the TOC to TAC ratio, its value was higher in the smokers group, but also no statistically significant associations were found with the dose of exposure to tobacco smoke (r=0.232, p=0.140). Close correlations between smoking and the levels of individual antioxidants were demonstrated by us and other authors, which suggests the involvement of other components in the antioxidant defense of blood in the case of exposure to smoking.

D. Ardalic, A. Stefanovic, J. Kotur-Stevuljevic et al., “The influence of maternal smoking habits before pregnancy and antioxidative supplementation during pregnancy on oxidative stress status in non-complicated pregnancy,” Advances in Clinical and Experimental Medicine, vol. 23, no. 4, pp. 575-583, 2014.

M. Chelchowska, J. Ambroszkiewicz, J. Gajewska, and T. Laskowska-Klita,” The effect of Tobacco smoking Turing pregnancy on plasma oxidant and antioxidant status in mother and newborn,” European Journal of Obstetrics & Gynecology and Reproductive Biology, vol. 155, pp. 132-136, 2011.

B. Ermis, R. Ors, A. Yildirim, A. Tastekin, F. Kardas, and F. Akcay, “Influence of smoking on maternal and neonatal serum malondialdehyde, superoxide dismutase, and glutathione peroxidase levels.,”Annals of Clinical & Laboratory Science, vol. 34, pp. 405-409, 2004.

U. Aydogan, E. Durmaz, C.M. Ercan et al., ”Effects of smoking during pregnancy on DNA damage and ROS level consequences in maternal and newborns,” Arhiv za Higjenu Rada i Toksikologiju, vol. 64, pp. 35-46, 2013.

L. Fayol, J.M. Gulian, C. Dalmasso, R. Calaf, U. Simeoni, and V. Millet, ”Antioxidant status of neonates exposed in utero to tobacco smoke,“ Biology of the Neonate, vol. 87, pp. 121-126, 2005.

F.S. Orhon, B. Ulukol, D. Kahya, B. Cengiz, S. Baskan, and S. Tezcan, “The influence of maternal smoking on maternal and newborn oxidant and antioxidant status,” European Journal of Pediatrics, vol. 168, pp. 975-981, 2009.

In our study, we showed that the relationships between OSI and NO level in both groups were on the border of statistical significance (previous: Table 3 and lines 199-200; actually Table 4 and lines 218-219 ) and positively correlate with iNOS and total oxidative activity in the smokers group (previous: Table 3 and 216-217; actually Table 4 and lines 240-242). And as suggested by the reviewer, linking OSI with nitric oxide are shown in the figures (Figure 3).  

At the same time, I would like to add, that the presented manuscript constitutes part of a broader project in which, in addition to the presented results, we are studying the adverse effects of tobacco smoking in pregnant women, and where the problem of the influence of NO activity and metabolism on the newborns’ condition will be the subject of wider analysis and discussion, and will be elaborated on in a separate future paper. We can initially confirm (unpublished data) that maternal NO is a significant predictor of newborns’ birth weight and length (p <0.001) in mothers who smoke.

In figures 1-4, the R2lines ought to be consistently marked as smoking or non-smoking, perhaps by making one of the lines consistently dashed, color coding, or labeling the line formula (R2S= …, R2N=…). 

The R2 lines for smokers and non-smokers have been appropriately labeled so they are clearly distinguished from each other.

Line 292-4, “In vivo studies showed that local exposure to nicotine is associated with decreased response resulting from the endothelial action of NO in human vasculature, suggesting that vasoconstriction may be the result of inhibited eNOS activity [40].” The report [40] says “nicotine administration was associated with a loss in sensitivity to bradykinin,” even in the presence of indomethacin that would block prostacyclin synthesis: this is suggestive but does not definitively relate the response to NO inhibition, because there is still a possibility of a third vasodilator, EDHF. A more conservative sentence might be, “In vivo studies showed that local exposure to nicotine is associated with decreased response to bradykinin in human vasculature, suggesting that vasoconstriction may be the result of inhibited eNOS activity [40].”

Corrected as suggested (line 316-318).

Line 300-1 is not clear to me: “An increase in inducible nitric oxide synthase activity and concentration may be a secondary process for ROS generation by tobacco smoke.” Does this mean that higher iNOS causes more ROS generation? Is it including NO as an ROS? Or should it say that higher iNOS results from ROS generation by tobacco smoke?

Corrected as the Reviewer suggested. The authors wanted to say that higher iNOS concentrations or higher iNOS activity may result from ROS generation by tobacco smoke (line 323-324).

Line 309-317: I would have supposed that the (fairly long lasting) decline in eNOS more than offset the (fairly transient) exogenous NO exposure plus iNOS activity.

Indeed, it can be assumed that the long-term decline in eNOS activity in smokers has a greater impact on reducing the amount of endogenous nitric oxide produced than transient exposure to exogenous NO from tobacco smoke. This valuable remark has been added to the text (line 341-343).

According to suggestion, the entire text has been carefully proofread, and errors have been corrected.

Reviewer 2 Report

In the manuscript IJERPH-387136 authors presented the results of a study investigating the relationship between nitric oxide and markers of oxidative stress/antioxidant defense in serum of N= 42 tobacco-smoking healthy pregnant women and of N = 57 non-smoking pregnant women (control group). Subjects were tested for nitric oxide (NO), endothelial (eNOS) and inducible (iNOS) nitric oxide synthase, total oxidant capacity (TOC) and total antioxidant capacity (TAC). NO, eNOS, and TAC serum concentrations were significantly lower (p<0.005) but iNOS (p<0.05), and TOC (p<0.001) were higher in smokers than in non-smokers. Multivariate regression analysis showed associations between NO concentration and eNOS, TAC, and smoking status in the whole group of patients. In the model estimated separately for smokers, the highest impact of eNOS was indicated for NO concentration (β=0.545; p=0.009). In the model of non-smokers, eNOS (β=0.291, p=0.030) and TAC (β=0.350; p=0.015) were significant for NO level. Based on the obtained evidences, authors concluded that: (i) smoking during pregnancy could exacerbate oxidative stress, impair the action of nitric oxide synthases, and adversely affect the balances of oxygen and nitrogen metabolism; and that (ii) relationships between NO concentrations and TAC in the studied women's blood can confirm the antioxidant nature of nitric oxide.

Overall, the manuscript is written in a clear exhaustive manner and with scientific soundness. The study, the objectives set, and the obtained results are very interesting and represent an advancement of the current state of knowledge. In summary, this is a well-written manuscript that deals with an interesting topic, addressed with a method that in some ways is innovative, that should be considered for publication on the International Journal of Environmental Research and Public Health. 

Author Response

Review Report Form 2

In the manuscript IJERPH-387136 authors presented the results of a study investigating the relationship between nitric oxide and markers of oxidative stress/antioxidant defense in serum of N= 42 tobacco-smoking healthy pregnant women and of N = 57 non-smoking pregnant women (control group). Subjects were tested for nitric oxide (NO), endothelial (eNOS) and inducible (iNOS) nitric oxide synthase, total oxidant capacity (TOC) and total antioxidant capacity (TAC). NO, eNOS, and TAC serum concentrations were significantly lower (p<0.005) but iNOS (p<0.05), and TOC (p<0.001) were higher in smokers than in non-smokers. Multivariate regression analysis showed associations between NO concentration and eNOS, TAC, and smoking status in the whole group of patients. In the model estimated separately for smokers, the highest impact of eNOS was indicated for NO concentration (β=0.545; p=0.009). In the model of non-smokers, eNOS (β=0.291, p=0.030) and TAC (β=0.350; p=0.015) were significant for NO level. Based on the obtained evidences, authors concluded that: (i) smoking during pregnancy could exacerbate oxidative stress, impair the action of nitric oxide synthases, and adversely affect the balances of oxygen and nitrogen metabolism; and that (ii) relationships between NO concentrations and TAC in the studied women's blood can confirm the antioxidant nature of nitric oxide.

Overall, the manuscript is written in a clear exhaustive manner and with scientific soundness. The study, the objectives set, and the obtained results are very interesting and represent an advancement of the current state of knowledge. In summary, this is a well-written manuscript that deals with an interesting topic, addressed with a method that in some ways is innovative, that should be considered for publication on the International Journal of Environmental Research and Public Health. 

Thank you very much for your  review of ours manuscript.

According to suggestion, the entire text has been carefully proofread, and errors have been corrected.

Reviewer 3 Report

This an excellent well designed, conducted and documented study.

Please conduct and report a review the literature on smoking and adverse effects on the progress of pregnancy and the fetus. For your sample please document from the charts any adverse effects during pregnancy and delivery and report the birthweights of the newborn.  

Author Response

Review Report Form 3

This an excellent well designed, conducted and documented study.

Please conduct and report a review the literature on smoking and adverse effects on the progress of pregnancy and the fetus For your sample please document from the charts any adverse effects during pregnancy and delivery and report the birthweights of the newborn.  

According to the Reviewer’s recommendation, we added more information about the adverse effect of smoking on the progress of pregnancy and the fetus Table 1, line 51-53).

At the same time, I would like to add that the presented manuscript constitutes part of a broader project in which, in addition to the presented results, we are studying the adverse effects of tobacco smoking in pregnant women, and where the problem of the influence of NO activity and metabolism on the newborns’ condition will be the subject of wider analysis and discussion, and will be elaborated in a separate future paper.

Nevertheless, detailed data on birth weight, body length, and Apgar scores are given in the Results, lines 179-184.

According to suggestion, the entire text has been carefully proofread, and errors have been corrected.

Round 2

Reviewer 3 Report

The authors have responded to all the suggestions of the reviewers. It is an excellent study. 

In the new text on line 81 hem should be heme.